# Connection 100—An Auto-Ethnography of My (Mystical) Connection Experiences †

Mike Sosteric 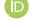

Faculty of Humanities and Social Sciences, Athabasca University, Athabasca, AB T9S 3A3, Canada;
mikes@athabascau.ca

† Let's face it, these hidden laws [of mysticism] are hidden, but they are only hidden by [your] own ignorance. And the word mystical is just arrived at through people's ignorance. There's nothing mystical about it, only that you're ignorant of what that entails—**George Harrison**.

**Abstract:** This paper provides an autoethnographic accounting and analysis of my own mystical experiences, called connection experiences in this paper. This account, which is structured around a description of my early experiences, attempts to weave together psychological, sociological, historical, and methodological themes into a coherent contribution that advances our understanding of connection experience. The paper includes an explication of the four stages of the research project that developed as a consequence of these experiences as well as an examination of the processes, tools (i.e., MediWiki), and emotional, psychological, professional, and scholarly challenges of collecting and analyzing the autoethnographic data of mystical experiences. The denouement of the paper is the presentation of a conceptual schema aimed at overcoming nomenclature confusion and providing a basis for description, analysis, and discussion of connection and connection experiences. The utility of the schema is demonstrated when it is used to provide a clear overview of my own connection experiences, and the connection experiences of others. In order to facilitate critical discussion of the conceptual framework, a glossary of terms developed and presented in this paper is provided at the end.

**Keywords:** mystical experience; indoctrination; autoethnography

## 1. Introduction

In 2003 I had a terrifying and traumatic mystical experience, what I later came to call a combination **Nadir Experience**[1] (because the experience was dark and terrifying) and **Clearing Experience**[2], so named because, as recounted below, the experience cleared fears that had been blocking me since my childhood. This experience was followed by a string of positive **Zenith Experiences**, which I define as positively felt mystical experiences. Taken together, these early experiences completely transformed my view of self, the world, the universe, the nature of reality, and my place in it. I made an early decision to engage in scholarly exploration of these experiences, what some are now calling "emergent phenomenology," but what I have come to call **Connection Experiences** (CE)[3] because they represent, depending on one's theoretical and ontological predilections, connection to either deeper neurological states (Newberg et al. 2001; Newberg and Waldman 2009; Persinger 1987, 2002; Rizzolatti et al. 2001) or to a wider (or deeper) realities of Consciousness (Dossey 2012), realities characterized by mystics variously as connection to an "incorruptible one" (Wisse 1990), an "ultimate reality (Happold 1963), a "love-fire" (Boehme 1912), an "inward light" (Kelly 1941), and by scholars and theologians as connection with vast and "ineffable realities (Stace 1960), a "numinous order" (Otto 1917), an "extended mind" (Jahn and Dunne 2009), a "real" real (Underhill 2002), a "supreme and ultimate reality" (Inge 2005, p. 8), and even a "vast intelligence," "marvellous order" (Einstein 1930) or "old one" (Martin and Ott 2013), as Einstein variously referred to the wider reality.

This paper is the initial analysis and report of a scholarly exploration and analysis of these experiences that has spanned nearly two decades and has proceeded in four rough stages, the Flow Stage (approx. 2003–2005), Analysis Stage (2006–2013), Grounding Stage (2013–ongoing), and finally, the Autoethnographic Stage (2021–ongoing) all of which I will recount in the main body of this before. Note that although I discuss the content of the connection experiences briefly, in this paper I focus less on the content of these experiences, a content which parallels closely the phenomenology of standard mystical experiences[4] and instead focus more on the psychological, emotional, and even political processes involved as I struggled to make sense of it all. I avoid engaging too much with the "mystical" elements not because these are unimportant. In fact, as Jones (2021) notes, these experiences hold personal, professional, scholarly significance and, I would argue, political significance. I avoid it because my connection experiences broadly mirror the phenomenology and outcomes of these experiences as reported in the psychological and theological literature, because I deal with this aspect of the experiences elsewhere (Sosteric 2016, 2018a), and because sociological issues of power, social class, gender, colonization, and contest, methodological issues of recording, analyzing, and reporting, and theoretical issues of conceptualization and nomenclature confusion seemed, particularly since these are not often treated in the literature on connection experiences, more important to discuss.

I would like to note before proceeding that this autoethnography, this qualitative research project, presented a peculiar challenge, that being how to convey the complex and powerful connection experiences that initiated this research project, while at the same time attending to a wide range of sociological, psychological, and historical issues, all within the context of a complex, and at times confusing (see my analysis of nomenclature confusion below) scholarly record. As a sociologist with a degree in psychology, and as someone who appreciates the power of both qualitative and multidisciplinary research, I am used to working with complex content, but this was a step above, requiring a synthesize of psychology, sociology, and history in the context of initially confusing experiences and a diffuse and confusing literature. The approach I chose to deal with this complexity is a braided approach to reporting. This is an approach used by historians (Donald 2012; Fischer 1976; Mullen 2019) and others, particularly those who write about Indigenous topics (Donald 2012), to structure writing in a way that can efficiently write complex phenomenon in a non-reductionist and decolonized fashion.[5] A braided approach weaves "analysis with narrative" (Mullen 2019, p. 384) and "juxtaposes diverse forms of texts" (Donald 2012, p. 53) as a way to ground and critically reveal complex content. A braided approach is similar to standard qualitative methodologies which tend to cast researchers as bricoleurs who deploy "strategies, methods, or empirical materials" as necessary to capture the area, even going so far as to invent new research tools if required (Denzin and Lincoln 1994, p. 2). "The product of the bricoleur's labor is a bricolage, a complex, dense, reflexive, collage like creation that represents the researcher's images, understandings, and interpretations of the world or phenomenon under analysis" (Denzin 2013, p. 3).

That is certainly what this paper is, a psychological, sociological, neurological, historical, and theoretical bricolage, a complex braid with my own connection experiences as the core around which the braid/collage is constructed. This is an appropriate methodology for a topic such as this one. Such an approach can provide a "tighter synthetic unity" (Fischer 1976, p. 119) to our understanding of the phenomenon, in this case connection experience. It can also provide a deeper understanding of the complex web of factors (e.g., social class, gender, culture, power) as these relate to connection experience. Nevertheless, the braided approach, the construction of a complex collage, can also lead to a "loss in analytic clarity" and coherence (Fischer 1976, p. 119). This loss of analytic clarity can leave one with an impression of theoretical and methodological chaos or, as one editor of an earlier draft of this article put it, the impression that this work is a "mix of 'free association' of very different kinds of theories and of conjectures." It can also, particularly in researchers who confine themselves to single methods, single theoretical traditions, single disciplines, and single epistemological frames, and who consequently expect a single narrative thread

that culminates in a small number of tight final conclusions, leave one with a nagging question—"Where is the research?" or "What's the contribution of this paper?"

The answer to this question is this: the research, the contribution, can be found not in a view of the trees, not in an isolated examination of connection experiences, but in a view that recognizes the surrounding forest—the fungi, the insects, the animals, the waterways, and so on—within which the trees of connection experience are embedded. The contribution is in the provision of understandings of connection experience that recognize an interpenetrating web of cultural, social-class, gender, and ethnic conglomerations all influencing and being influenced by the experiences themselves, and the various social and political interests involved. It is also to be found in the provision of general methods for approaching an auto-ethnography of connection experience, and in the provision of a neutral and agnostic conceptual framework for understanding, analyzing, and communicating about these ubiquitous human experiences (Sosteric 2018a). It's a complex and challenging bricolage that, owing to word limitations, may seem a bit thin at times. Hopefully, despite the limitations, the reader will view the paper more as a collection of pointers to areas needing additional research and attention. A measure of the success or failure of this paper will be the extent to which the reader comes away with a stronger interest an appreciation of connection experience, and a better understanding of the complex social, political, psychological, and perhaps historical issues relevant to our understanding. I will leave it up to the reader to judge and discuss the relative value of these contributions. With that said, let us begin.

## 2. Back Story

I was raised by a single parent, fire and brimstone Catholic mother who required my brother and I to attend Church on a weekly basis. I was never very happy with that. It was boring, for one, and even as a child I could recognize the contradictions. I could see that the Church as an institution, despite pretensions to ethical and spiritual superiority, was filled with meanness, hypocrisy and violence. My Catholic teachers were mean and violent. My Catholic mother, under the guise of discipline, was mean and violent. Our neighbours always seemed far below the minimum standards of Christian concern. For the entirety of my childhood, I lived in an uneasy relationship with the Church until one day I burned my hand by impatiently reaching into a hot oven to grab some cookies. My mom, instead of reaching out to hug and comfort me, grabbed my hand and screeched how hell would be much worse than the second-degree burns on my hand. It was at that point that I rejected the Church outright. Even though I was only eight, I could see that something was terribly wrong. I could not abide the psychological violence and so, blaming the Catholic Church for teaching my mom such fear-filled and hurtful nonsense, I left the faith. I continued to go to Church when forced but when I was old enough, I simply stopped going altogether.

Although I rejected the Catholic Church at an early age, I did not dismiss the possibility of spirituality outright and I did not stop my spiritual questing right away. I spent my adolescent years browsing the library and the local New Age bookstore that had recently popped up, looking at Theosophy, Eckankar, Astrology, Buddhism, Zen, and other outside-the-Catholic-mainstream spiritual offerings. To my young and uneducated mind, the writing was often obtuse and confusing, especially the elitist Theosophical stuff. As I passed into life as a young adult, I continued my search, bordering for many years on the fence between curious agnostic and disinterested/dissatisfied atheist. Then, years later, in my first-year Sociology course, I read the infamous words of Karl Marx—"Religion is the opiate of the masses". Going over all the things I had learned about society in that course, recalling my own traumatic experiences of religion, and the general dissatisfaction of my time in the book aisles of the New Age shop, I decided that Marx's assessment was certainly true. Religion was an institution that supported social control. Religion was a pacifier that prevented social change. God was the ideological and infantile delusion of an oppressed and infantile planet that Freud had made it out to be. The pins in the tumbler of my mind dropped neatly into place and I joyfully stepped through the doorway into the bright world

of atheist superiority. Subsequent to that, I did not want to have anything to do with God, religion, or spirituality. I focused on my studies, got my university degrees, and settled into a Sociological career. I was happy, satisfied, and sure the Catholic indoctrination had been cleared. I was proud of my accomplishment. I puffed my chest and ruffled my peacock feathers, proud of my born-again atheism. I knew the truth and the truth had set me free.

## 3. Connection Experience

Like many atheists, I was arrogant in my rejection of "the faith" and sure of the superiority of my perspective. But then one day, "it" happened. In 2003, at the age of 39, I had a powerful mystical/transcendent experience that frightened me almost to death. The experience, which I later came to call a Connection Experience (CE) (Sosteric 2018a), occurred in the first few minutes of a 1989 science fiction movie entitled the *Abyss* by director James Cameron. Since the past ten years has seen the redemption of cannabis, psilocybin, and other **Connection Supplements**[6] (CS) as viable psychiatric tools, I can admit that before the movie I helped myself, for the first time in decades, to a very small dose of cannabis. I sat down with my partner and, impelled by the small dose, immediately entered an altered state of consciousness far more potent than the tiny consumption of leaf should have induced. Sitting there watching the movie, I was confronted by the God of my Christian youth.[7]

You might expect that I experienced wonder, glory, and joy in this initial contact with God, but I did not. The experience was terrifying. For the first ten minutes of the film, I engaged in a "dialogue with God" (more of a monologue with terror) where I confronted horrible fears of rejection, judgment, and damnation. Sitting on the couch gawking at the television, I knew for certain that God had come to me in judgment and that I, as a consequence of all my sins, would be seen as unworthy and subsequently cast into a pit of eternal fire. It is hard to convey the existential terror of this dark night of the soul nadir experience[8] except perhaps to say that I was transfixed with terror. In that moment, I was a small child being confronted by an all-mighty parent who seemed intent on causing me eternal harm. Terrified, powerless, subjugated, defeated, worthless, useless, insignificant, inconsequential cosmic garbage are adjectives I could use to describe my feelings during this extremely unpleasant experience.

I have to say, this was not my first traumatic experience of God, judgment, and damnation. I had experienced this visceral horror show three or four times as a teenager and young adult. During these earlier experiences I would simply ride it out till the inevitable return to the "normal consciousness" of my day-to-day life, a state of consciousness more amenable, I have found, to the operation of psychological defences like repression, which I would promptly apply in order to get past the horror and trauma. However, this time was different. As I sat there anxiously reviewing all the "horrible" things I had done, as I gasped transfixed, awaiting the inevitable smiting, I got fed up. Tired of the fear, trauma, and emotional anguish, tired of feeling judged and worthless while sheepishly waiting for God to smack me down into the fire, I chose instead to stand up, metaphorically speaking. In my mind, I raised my fist in the air and gave God the middle finger. In the internal and infernal dialog that occurred in the few brief moments of this experience, I uttered a damnable heresy. I looked God square in the eye that night and I said that if "his" idea of love and acceptance involved eternal fire for those children who did not live up to "his" standards and expectations, then "he" could take his pathetic creation and fuck right off because I did not want anything to do with it anymore. If he wanted to throw me into the fire for my insolence, fine, do it. After this internal emotional outburst, I sat back, took a few deep breaths and waited for the lightning bolt to strike me down. However, the bolt never came, at least not in the form that I was expecting. So, traumatized, I watched the rest of the movie, after which, still shaking from the frightening experience, I went to bed.

The next day I woke up still traumatized but surprised that God had not murdered me in my sleep. As I reflected upon my experience, I considered the existence of God, but rejected that. Surely if God existed, at least in the Catholic form of my indoctrinated youth,

"he" would have done something about my brash disrespect. Yet, surprisingly, there was only silence. If God existed, God seemed unconcerned. At the very least, it was clear that if a God did exist, this was not the abusive and violent patriarch that I had been taught to expect. Whether God existed or not did not matter at that point. The thing for me was the intense relief that set in as I gradually came to suspect that the fears seeded by Catholic priests in the deep dark recesses of my mind were completely unfounded. If God had not smacked me down after that insolent outburst, obviously I had nothing to fear.

### 3.1. Grounding the Event

I went through the first post-event day slightly dazed and with lingering tendrils of fear and trauma still pressing against the borders of my mind; eventually, however, these dissipated. The next day I felt better and the day after that I was cognitively and emotionally back to normal. At that point, had nothing else occurred, I would have pushed the events aside and continued along with my traditional sociological interests.

To be sure, I did think about the events and I did make some observations. I observed, for one, the intensity in the *flow* of ideas that invaded my internal monologue that night. They came spewing as if through a pressured fire hose. It was, frankly, hard to handle. I also observed the existential terror that came with these ideas. Interesting, I also observed that the only thing that relieved me, the only thing that "saved me" from the terror and pressure was facing directly into the flow, accepting the ideas were there, and dismissing them as theological/cosmological nonsense. In addition to observing the intensity, I also observed that childhood indoctrination had instilled recalcitrant ideas (archetypes, as Jung (1980) might say) and powerful fears deep in the unconscious recesses of my mind. It was interesting that even though I had consciously rejected the root doctrines years earlier, these fears lay dormant in my unconscious mind, waiting to be triggered and amplified by the small dose of cannabis sativa. Recognizing that the fears were there, I wondered if these fears could be triggered again. Finally, I wondered briefly about the source of the flow. I concluded initially, that cannabis has activated unconscious areas of my neurological brain, areas that I had suppressed years ago. Thinking this was probably the end of it, I was ready to settle back into normal consciousness and normal daily routines.

### 3.2. The Flow

Settling back into normal was, however, not in the cards. Just a few days after I uttered "the heresy," I felt an inexplicably intense internal pressure to write. Giving in to this pressure, I sat down in front of my computer screen and with touch typist speed began to inexplicably flow words onto the page like a fire hose flows water onto a fire. In the weeks and months that followed, a wealth of *raw* content churned from my fingers, sometimes on topics I knew something about, and at other times on topics I had never considered before.[9] In the span of three or four days, I wrote a creation story, two poems portraying apocalyptic eschatological denouements to Earth's evolutionary unfolding, and an allegorical representation of the psychological processes of "awakening." Surprised by the sudden burst of creativity, I continued with this process, engaging in what I later came to call daily **Connection Practice**[10] by sitting down for an hour a day. I did this for several months until finally I had to sit back and reflect. As content continued to churn with no sign of abatement, I realized I had a decision to make. Should I continue in this daily practice, spewing the raw gusher onto the page, or should I stem the stream of consciousness and return to my "normal" pre-event baseline?

It was not an easy decision to make. There were several reasons to put this all aside. One reason was that I was confused and a little distracted from my normal routines. Typing the flow on paper was easy enough, but understanding what was going on was not. Like a coxswain in a cockleshell cast into the middle of a vast ocean, I was lost. I had no idea where the ideas and the visions were coming from, I had no idea why there was such pressure to write things down, and I could not explain the copious flow. I had no idea what to make of it all, though I did find out many years later that copious

flow was a characteristic of the mystic's CE ([Harmless 2008](#)). Another reason to put it all aside was that I experienced considerable cognitive dissonance. At the point of these experiences, I was an atheist that had rejected religion and human spirituality as atavistic nonsense. The obvious spiritual, theological, cosmological content of much of the content represented a fundamental challenge to an atheist world-view I had formerly considered secular gospel. Besides confusion and cognitive dissonance, a third good reason for putting it all aside was that I consistently confronted additional fears that, while not as powerful as those I experienced during my initial clearing experience, nevertheless made me reluctant to pursue further exploration. One particularly troublesome fear was the fear of being overwhelmed. In the early days, ideas gushed so fast that I struggled to get them on paper. At times I feared I might be overwhelmed by the intensity. Finally, I also worried about my sanity and whether or not I could stay grounded over the short and long term. The ideas that came with the flow, though mostly reasonable, insightful, and enlightening, were at times quite outside the boundaries of acceptable mainstream thought. As I discovered later, these fears of losing control and losing ground are not uncommon. Several individuals I have spoken to over the years have indicated the presence of these fears in their CEs.[11]

It should be noted that the fear of losing control and going mad are not irrational fears. Though rare, it is absolutely possible to be overwhelmed and lose control of the flow, perhaps permanently. At least one case study exists ([Schreber 2000](#)) and I have known several individuals who struggled to maintain ground and stay functional in normal reality once they began to experience regular connection. One individual that crossed my path in fact lost this struggle. Connection was so overwhelming that their bodily ego completely dissolved as they took on the eschatological mission and identity of every famous spiritual person that ever existed, reporting themselves to be the reincarnation of Jesus, Maitreya, Allah, Zoroaster, and more. Although I lost contact with this individual, I do know they had sought psychiatric assistance and consequently were placed on anti-psychotic medication in an attempt to shut down the out-of-control flow that had smashed through the boundaries of their bodily ego and swept their sanity away.

While we are still on the topic of fear, I would like to note that the original fear of a violent and judgmental patriarch and the associated feelings of unworthiness did not completely disappear after the initial nadir experience. These fears periodically resurfaced during my early connection practice. The occurrences were not nearly as intense as the initial dark night experience, however, and I quickly learned that I could simply press the thoughts and feelings away by practising what I later came to call **Flow Control**.[12] Over time, the fears became less salient until eventually, probably six or eight months into my regular connection practice, all remaining neurological tendrils had been erased and the fears no longer appeared.

To summarize, confusion, disorientation, cognitive dissonance, and various fears presented a barrier to proceeding with research.

Despite all that, however, there were compelling personal and professional reasons to continue. The creative enhancement alone, the sudden ability to write poems, parables, and allegories no matter how artistically immature they may have been, was enough to signal the worth of an investigation. However, there was more. The theological, cosmological, and psychological content of the experiences, content which I am still processing and trying to understand, were effusive, wide-ranging, and fascinating. There was also considerable cognitive and emotional benefit. As others have observed, connection brings cognitive enhancement ([Bucke 2009](#); [Hanes 2012](#); [Ikbal 2000](#); [Lydon 1982](#)), psychological and emotional insight or epiphany ([Bidney 2004](#); [Miller and Baca 2001](#); [White 2004](#)), expansion of empathy and love ([Parish 1999](#)), to name a few positive benefits.[13] I can humbly attest to all these things. Regular connection practice, when properly grounded and free of fear and oppressive and restrictive ideology, does lead to cognitive and emotional enhancements.

There were also professional reasons which encouraged me to move forward. Particularly salient was the dearth of sociological research on the topic. A brief survey of the literature in the first few months of the research program revealed that sociologists

had given short shift to this phenomenon, which surprised me considering its obvious significance. Later, during a more focused review of the literature, I realized just how short was their shrift. Despite the fact that there had been some discussion of mysticism, mystical experience, and other anomalous phenomenon during sociology's birth and adolescent stages (Garrett 1975; Wach 1947), subsequent to that initial interest, mysticism and mystical experience has been almost completely ignored (Johansson 2022). Save for a handful of studies during sociology's adolescent stage (Bourque 1969; Bourque and Back 1968; Furfey 1940; Robertson 1975; Sturzo 1942), and a nascent interest re-emerging only very recently (Winchester and Pagis 2022), sociologists have focused exclusively and unfortunately on the ecclesiastical side of things, with a smattering of cult investigations on the side (Bainbridge and Stark 1980; Wallis 1976). This dismissal of mystical experience has left sociologists' understanding of human spirituality a poorly drawn caricature, or as Bourque (1969, p. 151) charges, "highly stylized" and "simplistic . . . " To be clear, the conflux of my sociological training, my spiritual exploration, and the startling dearth of sociological interest, presented a unique opportunity—one that was hard to put aside.

Eventually, about eight months into the process, I did decide to move forward. Assessing the situation, I eventually determined the flow would not be a problem. I was a fast typist and I could get ideas down without them building to the point of overflow. Although fear remained an issue for some time, as my control over the flow grew through practice, lingering fears of judgment and damnation, fears that had previously caused interruptions and distortions in the flow, slowly dissipated and finally vanished altogether. As for the possibility of madness, I was able to create emotional, psychological, and physical boundaries around the experiences. These boundaries allowed me to preserve my normal life and function despite the odd explorations. Finally, I was also able to avoid censure by simply keeping my research and exploration private. I knew I would have to talk about it eventually, and I knew this would be a challenge, but I also knew that if I was to avoid ridicule and censure, I needed a sensible framework to present. I decided I would simply keep silent until I was confident I understood and could explain in sensible terms.

## 4. Methodology

With the decision to move forward, the question of methodology arose. As a trained social scientist, I knew I had to formulate a specific purpose, develop specific research questions, collect data, and eventually analyze, interpret, and share (Chang and Boyd 2011). I had done ethnographic research in the past, so I had a decent overall sense of it. My research purpose was to explore and try and understand these mystical CEs. As for specific research questions, at the beginning these were basic and broad. What was the source of these experiences? What were they about? Why had I not learned about these things in Catholic Church? Where did the fear and confusion come from? What was the connection between childhood trauma and religious indoctrination, disconnection, and reconnection? Why, after clearing the fear, did the flow begin? What other factors undermined or supported CEs? Why, despite initial interest, had sociologists ignored the phenomenon?

### 4.1. Stage One: Flow (2003–2005)

Once I had made the decision to move forward, my thoughts turned to data collection. In this regard, I knew I had to record as much as possible, so that is what I did. Data collection for this project occurred in two stages. **Stage one**, which we might call the **Flow Stage**, proceeded during the first two years of daily exploration. During this period, I engaged in disciplined and daily connection practice, waking early in the morning before everybody else, sitting down at my computer, and recording the flow that occurred onto the page as crisply and transparently as I could manage. My approach during this period was simple. I kept my critical mind off and my fingers moving, setting down content without thinking about it, as fast as I could. During the flow period, I did not vet the content in any way. I felt if I did, I would be tempted either to ignore certain concepts or

ideas which were to "out there" or I would be tempted to modify the flow to fit into more "reasonable" frameworks, as defined by the filters of my own experience and ingrained scholarly expectations. To avoid that, to secure as accurate and transparent a record as I could, I just threw the ideas on paper as fast as they came out.

Overall, I think I managed a decent and transparent representation. My ability to type rapidly meant I was able to get all the ideas on paper. The density of the flow ensured I had little time to reflect on or second-guess the flow as it passed consciousness and onto the page. I am reasonably confident when I say that the initial flow was not unduly influenced by passage through the *conscious* aspects of my mind. I cannot say the same for any deep unconscious influences that might have tinted the flow, and I do believe these unconscious influences exist, specifically in the ideas and archetypes I had received via the religious and secular myths and stories of my youth, adolescence, and early adulthood. However, I did not worry about these internal influences during this stage—I just flowed the information onto the page.

In addition to not vetting the ideas as they flowed, during stage one of data collection I did not consult the scholarly or popular literature to any significant degree. As noted earlier, I did have some adolescent experiences with the popular spiritual/new age literature, and I had glanced at the scholarly literature when these experiences began, but what I had found in both cases did not help me understand what was happening. Scholarly literature was often simply descriptive. In addition, theoretical understandings and explanations seemed, in the light provided by my own experiences, caricatured, two-dimensional, simplistic, and sexist.[14] Popular literature was even less help. That literature ranged from benign but theoretically useless "conversations with God" to channelled eschatological analysis of the end-times unfolding by disincarnate entities from the Pleiades, to paranoid and racist tracts about lizard people and alien invaders. Given the unsatisfying nature of my initial explorations, I thought it best to avoid the literature and just let the process unfold as organically as possible.

*4.2. Stage Two: Analysis (2006–2013)*

The flow stage went on for about two years. At a certain point, the flow began to diminish and I began to feel a desire to examine, analyze, revise, clarify, and ground the information that I had "received." Thus began stage two of the research process, the analysis stage. During this stage, I began to go through everything that I had put on paper, editing and modifying, clarifying and expanding concepts and ideas, and generally trying to make sense of it all. I initially started doing this analysis and revision in the actual documents but realized I was muddying and destroying the record. So, I made copies of what was left of the original documents for preservation. In addition, in order to track the process and evolution of ideas, I also installed MediaWiki software in 2007, provided online access, and then used that software to record concepts and ideas directly—as a sort of academic notebook,[15] which I call the SpiritWiki. MediaWiki software is ideal for this. The software records the date, time, and substance of every edit that you make and is a perfect tool for recording the conceptual ebb and flow of an autoethnographic project.

As amazing as MediaWiki software is, the record I have is not perfect. It was two years into the project before I got the idea to use a wiki to record the process. Furthermore, when I created the SpiritWiki, as I call it, I did not systematically proceed through each resource. I continued collection of autobiographical data and I entered and modified concepts as I focused on this or that stream. Consequently, the date of creation does not correspond to the date when ideas where first formulated, at least in the early days; they correspond more to focus dates, periods when I focused on this or that resource or idea. More recent entries can be dated more precisely because I have gotten much better at entering concepts as they percolate up; however, even that is not completely accurate.

Speaking of imperfect records, I should note, at a certain point I did in fact succumb to the temptation to "cleanse the flow." As I progressed through Stage Two Analysis, I became concerned that some of the early formulations might be too distorted, and consequently too

dangerous, for a public record. I was also concerned what other scholars might think. I had spoken to one colleague about my experiences and he called me a space cadet. Another had read my original paper on the tarot deck but advised me to sanitize out any discussion of spirituality and connection. From this, I got the clear message that I should not be talking about these things because if I did, people would think I was crazy. This made me concerned for my livelihood. Though not a big fish like Rupert Sheldrake, I had to wonder, would I be excommunicated for divergent thinking as he had been (Freeman 2005)? Consequently, I chose to sanitize the record by deleting the first two years of revisions. I do not regret the decision to cleanse the public record, but had I the opportunity to engage in this sort of research project again, I would keep a private copy of the original database simply to preserve a more complete record of the project.

I should also note that MediaWiki software provides functionality that allows one to annotate entries and modifications. Unfortunately, I did not use this facility, mostly because I did not conceive of the need for it. There were many routine edits (spelling, grammar, etc.) that did not seem to require annotation and I thought that the evolution of the flow would be relatively straightforward to track simply be examining the larger edits. As I have been preparing this paper however, I have come to view the failure to annotate edits as a methodological error and would certainly encourage anyone else using MediaWiki software as an autoethnographic tool to consider making full use of its annotation capabilities.

Despite the limitations of my use of the MediaWiki software, it has proven to be a valuable record. Having all the concepts and ideas online, accessible, searchable, and easily linkable made it much easier to organize, systemize, and analyze the content. The SpiritWiki, for example, was a first step towards providing myself with mind maps of the concepts that percolated up during connection practice, which in turn helped me systematize the data. It also allowed me to identify problems in the scholarly corpus, like nomenclature confusion, which is an emerging concern in the literature (Johansson 2022) and which I will speak about more below. Finally, the SpiritWiki has also provided valuable reminders on the timing of the project, insight into how my thinking has evolved, and windows into the struggles I encountered with various psychological, emotional, conceptual, and ideological issues.

As with the flow stage, during the analysis stage I also avoided the scholarly and popular literature. I did this for the same reasons noted earlier. This proved to be a wise decision since later on in the grounding stage of the project, a stage where I did in fact dive into the scholarly and popular literature, I began to see elite interference in the exoteric and esoteric spiritual narratives of this planet, an influence that I may have been susceptible to had I not already built a strong foundation for my own understanding. A particularly potent example of this influence is provided by the Western Tarot, which is an extremely sophisticated ideological system obscured inside a fanciful and attractive spiritual narrative. The Tarot was created by Freemasons with the specific and express purpose of lubricating the transition from Feudalism to Capitalism (Decker et al. 1996; Sosteric 2014). To be clear, it is a tool for indoctrination, a perfect example of how elites colonize and manipulate this planet's spiritual narratives, and an example of how easily their constructed ideology is taken up not only by a population of naive spiritual seekers, but academics as well (Sosteric 2014).

Stage two of this research went on for about eight years, from about 2005 to 2013. During this time, I continued with rigorous and disciplined daily connection practice. This is a long time to engage in daily connection practice and analysis, but it was necessary for both cognitive and emotional reasons.

Cognitively, the corpus that eventually coalesced was complex, as is often the case (Harmless 2008). Sorting that out took a lot of time. There was also the unconscious influence of archetypes on the flow of ideas. These archetypes, mostly sourced from elite spiritual narratives, and discussed in a bit more detail below, have, I think, a heavy influence on how we receive and transmit internal knowledge flows of the kind that occur during connection events. It took a while to sort the influence of these archetypes out to

the point where I felt that the ubiquitous elite impositions were not unduly influencing the reception, analysis, and eventual presentation of materials.

Emotionally, daily connection practice gradually revealed various "things" I needed to resolve, like childhood trauma, maladaptive emotional and behavioural responses, and unconscious misogyny, to name a few. Put another way, analyzing and grounding the materials required me to confront and heal certain emotional and psychological damage that had accrued during my childhood, adolescence, and early adulthood experiences, particular as this impinged on the health of my bodily ego, and which made me, and perhaps many others, susceptible to what I call **connection pathologies**[16] like **ego inflation**, which is an inflation of ego leading to delusions of spiritual grandeur. To be clear, getting to a place where I could process and assess materials required cognitive development, cognitive effort particularly focused on understanding the nature of elite interference, and emotional and psychological healing, all of which took lots of time.

Note, I do not think the requirement for analysis and healing are necessary parts of the process. As I have come to understand, the need to heal arises as a consequence of damage done by an intentionally designed and incredibly destructive Toxic Socialization (Sosteric and Ratkovic 2016) process characterized by violence, neglect of our Seven Essential Needs (Sosteric and Ratkovic 2022), chaos (in the home environment), indoctrination, and distortion/destruction of healthy family attachments. If this toxic socialization process was not imposed, we would not need to sort through confusion, we would not need to heal from damage incurred, and establishing more permanent and persistent connections would be less troublesome and easier to accomplish.

*4.3. Stage Three: Grounding*

Eventually, I felt I did start to make sense of things. Insights delivered via connection facilitated healing of childhood and adolescent trauma which over the course of time strengthened my bodily ego to the point where I believe I could handle the materials in a more objective and critical fashion. As I healed, as I put all the concepts into the SpiritWiki. This entry into the SpiritWiki facilitated analysis and understanding. As I thought about and analyzed the corpus, consistent and sensible conceptualizations began to emerge. At the point where I began to feel comfortable that I had established an internally consistent frame, I began to move into **Stage Three** of the project, the **grounding stage**. In this stage, I attempt to ground the information that I had been flowing into the scholarly corpus. I began reading the literature, adding (in the SpiritWiki) citations to my own concepts, the concept of others, and generally building up the scholarly foundation. This process led to my first "discovery," which was that elite actors had colonized the spiritual corpus of this planet. Evidence in this vein came from my investigation of the Tarot deck, a tool often presented as an ancient fountain of spiritual wisdom. However, as I discovered, the Tarot as we know and use it today is not a spiritual tool at all. It is in fact a propaganda device, part of what Decker, Depaulis and Dummet (Decker et al. 1996, p. 52) call the "most successful propaganda campaign ever launched: not by a very long way the most important, but the most completely successful. An entire false history, and false interpretation, of the Tarot pack was concocted by the occultists; and it is all but universally believed." The tarot itself is an elite constructed discourse invented by the emerging industrial bourgeoisie during the Industrial Revolution for the purpose of lubricating the transition from Feudalism to Industrial Capitalism (Sosteric 2014).

The initial discovery that elites had created the modern Tarot led me to discover that elite interference in spiritual narratives extends back centuries. An important example is Zoroastrianism. Zoroaster was a mystic who had received direct communications from God and the spirits (Boyce 2001, p. 17). Following these revelations, Zoroaster's teachings were spread word of mouth for over a century before finally being written down, at the behest of the Persian autocrat Aradashir, around 300 A.D. by Tanser, an elite Sassanian priest. Tanser, through appropriation, corruption of the teachings, and raw, suppressive

violence, colonized Zoroastrian teachings and created, under the direct authority of his regent, the Zoroastrian church (Boyce 2001, p. 3). As Mary Boyce notes...

> "...in place of the former fraternity of regional communities, a single Zoroastrian church was created under the direct and authoritarian control of Persia; and together with this went the establishment of a single canon of Avestan text, approved and authorized by Tanser ... Tanser set about his business and selected one tradition and left the rest out of the canon. And he issued this decree: The interpretation of all the teachings of the Mazda-worshipping religion is our responsibility". (Boyce 2001, p. 103)

It is notable that the Zoroastrian framework, which was created as a consequence of elite colonization of the "fraternity of ... communities" has penetrated deeply into almost every religious and cultural tradition on the planet and has had, as Boyce correctly notes, "more influence on humankind, directly and indirectly, than any other single faith" (Boyce 2001, p. 1). You find this framework in traditional religions like Christianity, Islam, and (despite Judaism predating Zoroastrianism) Judaism as well, where it was inserted via the Kabbalah teachings that emerged during the European Middle Ages (Dan 2006), in secular cultural productions like Star Wars and Harry Potter (Sosteric forthcoming), and in philosophy and critical thought of folks like Kant, Schelling, and Hegel (Wach 1947).

Why would the autocrat Ardashir claim interpretive superiority, reduce the Zoroastrian faith to a single cannon, and violently subdue competing understandings? For the same reasons Freemasons colonized an innocent pack of cards, Constantine co-opted Christianity (Sosteric 2020b), King James provided his own translation of the Bible, a Russian noblewomen Madam Blavatsky created the New Age movement, modern day capitalists co-opted the same new age movement (Carrette and King 2008)—to bend spiritual narratives to serve elite agendas. Ardashir took Zoroaster's teachings, teachings which had anti-elite elements (Dhalla 1938), twisted those teachings to suit elite rule, and then used the Zoroastrian religion he created to consolidate control of the spiritual narrative in a way that heightened his power over the people. As concisely as possible, he used it to build a discourse (Foucault 2012a, 2012b; McHoul and Grace 1993) or hegemonic frame (Hoare and Sperber 2016) that provided religious archetypes, cultural beliefs, and values all of which governed how people act, perceive, and feel, what they perceive as common sense, and through which people willingly submit to autocratic authority, and willingly insert themselves into a hierarchical, exclusionary, and imperialist cultural dynamics. I discuss the details of the archetypal framework created by Tanser (Sosteric forthcoming), as well as the nature and significance of these frameworks (Sosteric 2020a, 2021), elsewhere.

The discovery of elite interference in the spiritual fabric of this planet led to an important question which was, "what is it about human spirituality that led, and leads, elites to spend so much time and effort co-opting the artifacts and constructing their agenda-serving narratives?" The answer to that, I think, is simple. Connection experiences, mystical experiences, are extremely powerful experiences and most people have them (Sosteric 2018a). CEs are consistently linked to improvements in psychological and emotional health and well being (Bien 2004; Maslow 1964). More to the point, connection experiences can lead to dramatic personal and political insights (Harvey 1998), thereby facilitating what I call a "turn to the left" (Sosteric 2018b). A healthy, insightful, progressive population made so by regularly connection experience is an obvious threat to the status quo. Given that elite exploitation of the masses requires that these masses remain sick, ignorant, and in tacit or active support of the System, any powerful experience that causes healing, insight, transformation, and progressive political shifting, and any doctrine or dogma that encourages and facilitates said experiences, is a serious threat to their system of accumulation. Therefore, they co-opt and control.

My realization that "mystical" experiences were common and that they potentially led to transformative and progressive change, coupled with the realization that elites spend a lot of time and money messing with the narratives in order to suppress and confuse awareness, led me to theorize, after Ruyle (1975), that religion is not only a community projection

(Durkheim 1995), not only an opiate used to anaesthetize the masses (Marx 1970), and not merely an institutional wrapper for mystical experiences (Wach 1947), but also, at times, a sophisticated **Ideological Institution**, a "special instrument[s] of…thought control…staffed and/or controlled" by those who benefit from and therefore seek to, consciously and with considerable vigour, maintain systems that provide them with "special privileges and wealth" (Ruyle 1975, p. 11). Ideological institutions (the Catholic Church, the Freemason's lodge, etc.) are created to colonize and control spiritual discourse for the express purpose of manipulating and controlling not only the masses, but the elites themselves (Sosteric 2014). This is hardly a unique insight, and sociologists continue to point out how religion and the agents that work within these institutions do not work as "change agents working towards systemic reform" but as lubricants that help "keep the economic system running at the micro-level" (Cadge and Skaggs 2019, para. 8).

Of course, this is not to say that religion is nothing but elite interference, that it is only an ideological institution, or that it is all about power. Although the ideological and social class aspects of religion are important, particularly for the analysis of mystical experience in this paper, religion is obviously more than just a hegemonic tool for elites. As Ninian Smart suggests, religion is a complex of social, ritual, experiential (e.g., mystical experience), narrative/mythic, doctrinal, ethical, and material dimensions with an array of complex functions that include, among other things, servicing the community, providing emotional support, and meeting the spiritual and philosophical needs of the people (Smart 1973, 1992). In addition, as a mystic, someone Jones (2021) defines simply as an individual who engages in regular connection practice, I would and have argue that religion itself, and even the much maligned bible, can be quite transformative and empowering—the very anti-thesis of elite ideology and control (Sosteric 2018b). The Catholic Church, for example, was founded on the progressive teachings of Jesus Christ (Sosteric 2020b). Finally, despite the elite aspects of Catholicism, progressive teachings continue to influence the Catholic religion, its practices and teachings, particularly through progressive Liberation Theology (De La Torre 2013) and the efforts to create a "Black Theology" centred on the experiences of African Americans (Clark 2012). And note, Catholicism is not the only complex religious institution that is about more than elite power and control. Wicca is very much an effort to assert women's power via the rehabilitation of Earth based pagan religions (Starhawk 2011). Wicca is clearly outside of the elite sphere of influence. Clearly, religion is about more than elite power and control.

Nevertheless, it is also important and appropriate for a sociologist like me to focus on elite colonization of religious institutions and spiritual narratives. This is particularly true since despite being theoretically and empirically significant, this elite interference has hardly been addressed in sociology which has, until quite recently, maintained a tight ecclesiastic focus in its study of religion and spirituality (Perry 2020). The closest sociologists have come is probably the work of Foucault who spent a lot of time discussing the historical emergence and powerful significance of elite colonized discourse, e.g., in the legal system and in psychiatry. However, Foucault never conducted the same analysis on religion, perhaps thinking, like other secularists, that it was dying and therefore unimportant. I do think that given the significance of connection experiences and the obvious interest elites have in controlling religious institutions and spiritual narratives, there are important questions to address, like why the interest and what is the nature of the interference. I explore these questions elsewhere but to summarize, I would argue that elites have an interest in religious institutions and spiritual narratives because human spirituality, specifically connection and connection experiences, are not only a ubiquitous (Sosteric 2018a) and powerful threat to the status quo (Sosteric 2018b, 2020b) but also a powerful tool of hegemonic control (Sosteric 2014), They therefore need to contain and control religious institutions and spiritual narratives (Jantzen 1995), which they do to very great effect. I would also point out the archetypal nature of elite interference. When they get control of a religion or a spiritual narrative, they mess with the archetypes of that narrative (Sosteric 2014, 2020a, forthcoming). They modify them, as they did with their Tarot deck

(Sosteric 2014), the Zoroastrian faith (Boyce 2001) and the grass-roots Catholic Church (Sosteric 2020b), to suit their ideological needs.

I do want to spend too much time on this braid. I mention these things briefly because, as a sociologist, elite influence was one of the things that I attended to and because, as a sociologist, I think it is important to attend to this. In fact, I think that psychologist, historian, sociologist, or lay person, any grounded and sensible understanding of religion and connection experience will have to address not only the veridical nature of the experiences, but the powerful influence the elite's have had on the religious institutions and spiritual narratives of this planet. I would certainly encourage further psychological, sociological, and historical research along these lines.

### 4.3.1. Nomenclature/Theoretical Confusion

Elite interference in the spiritual/archetypal narratives of this planet is not the only problem I discovered as I compared my experiences and analysis with what I found in the literature. Particularly consternating, I found, is the proliferation of terms used to describe spiritual events and the conceptual and theoretical confusion, the **Nomenclature Confusion**, that ensues as a consequence. This confusion is most obvious in relation to the core mystical experience, which is referred to by a bewildering plethora of terms like flow experience (Csikszentmihalyi 1975), restorative experiences (Williams and Harvey 2001), beatific visions (Zaehner 1969) unity experiences (Kacela 2006), peak, plateau, and transcendent experiences (Maslow 1964, 1969), pure consciousness events (Forman 1986), perfect contemplation (St. Teresa of Avila 2012), trance (Stevens 1983), ascension (Harmless 2008), and satori (Suzuki 1994) to name just a few. The confusion is exacerbated by the fact that ontological, epistemological, spiritual, and even social class backgrounds (Bourque and Back 1971) can colour one's interpretation of the experience and subsequent selection and definitions of terms. For example, an atheist calls a connection experience a "peak experience" while a theist calls it a "union with God." In addition, confusion can be exacerbated because clear distinctions are not always made between the phenomenology of the experience, what it feels like, and the outcomes of these events, like healing, union, enlightenment or even psychosis.

Is nomenclature confusion a problem? One could argue, as Katz (1978, 1983) did, that connection experiences vary by the culture, perspective, and experience of the person having them and therefore this confusion is probably inevitable. Cultural variability is certainly a thing and since it is, we can expect different people from different cultures and different religious systems to use different terms to describe their various experiences. Others disagree, seeing nomenclature confusion as a problem (Dossey 1989; Johansson 2022; Parsons 1999), and for good reason. On the one hand, and most obviously, nomenclature confusion muddies conceptualization, convolutes attempts to define, confuses analysis, and makes communicating about the phenomenon in a way that leads to advanced in understanding a challenge. This difficulty is exemplified by the fact that despite over a century of study, "there is still no agreed-upon scholarly definition of 'mysticism' or 'mystical experience'" (Jones 2021, pp. 5–6) In fact, it is complete definitional chaos, with authors each constructing and theoretically grounding their own definition on the fly (Jones 2021). On the other hand, more subtly and precisely because this confusion allows authors to construct their own idiosyncratic perspectives, nomenclature confusion allows for the penetration of patriarchal, neo-liberal, colonial, and other forms of bias into the discourse by allowing individuals to construct their own definitions as needed, thereby giving them the space to arbitrarily exclude certain experiences that do not fit their particular bias. For example, one of the early giants in this field, William Stace, rejected raptures, trances, voices, visions, "hyperemotionalism," sexual feelings, and other sensuous/kataphatic female type experiences common, or so he felt, among females as not genuine or not examples of "highest expression" of a mystic's consciousness (Stace 1960, p. 47). Stace can speak for himself here.

But there can be no doubt that the abnormal bodily states which mystics call rapture or trance do sometimes occur. They are mentioned here as being of interest, but the point to be made is that they are accidental accompaniments of mystical consciousness, by no means universal or necessary. They occur among the more emotional and hysterical mystics and not among those of the more calm, serene, and intellectual types. They cannot therefore be regarded as belonging to the universal core of mystical experiences". (Stace 1960, pp. 52–53)

Stace labels mystical experience with emotive and sensuous elements as soft, hysterical, unimportant, lacking in balance and judgment, and devoid of critical sensibility. Stace even arrogantly dismisses one of the most famous female mystics of all time, St. Teresa, because she "was not an intellectual as Eckhart [or he] was, and not capable of much analytical or philosophical thinking" (Stace 1960, p. 49). Not to put too fine a point on it, but it is easy to make these exclusions in a field characterized by terminological inconsistency and conceptual confusion.

For the above reasons, the impediments to communication and understanding and the space it leaves for the importation of bias, nomenclature confusion is a problem.

### 4.3.2. Conceptualization, Operationalization, and Theorization

In order to help alleviate this problem, a phenomenologically and ontologically agnostic schema, a schema that can handle both theist and atheist orientations, that incorporates a wide array of experience, that does not lend itself to bias, and that can fully operationalize connection experiences, was developed during the course of this project. The roots of the schema are the concepts of connection and connection experience. These terms replace the term mysticism and the various phrases used to describe mystical type experiences (peak experiences, transcendent experiences, union experiences, flow experiences, etc.). Thus, when somebody has a mystical, transcendent, peak experience, etc., we say they are having a connection experience. How do we know they are having a connection experience? We know they are having a connection experience, or we know we are having a connection experience, because of the identifiable phenomenology of the experience which feels different than "normal" consciousness and which always comes with a feeling of being connected to something more than one's normal reality or every-day self (Jones 2021). To be clear, when we connect and have a connection experience, we know differently, we feel differently, and we sense we are connected to something that is different than, bigger than, more than our day-to-day reality and individualized, atomized ego.

Why the terms connection and connection experience and not something else? This for at least three reasons. Number one, the terms are completely consistent with the basic phenomenology of these experiences which consistently produce a "sense of connection of apparently separate realities" or of connection to more than one's own individual, atomized self (Jones 2021).

Number two, the terms are usefully agnostic. When one calls a connection experience a mystical experience, a transcendent experience, a peak experience, etc., one is implying certain ontological perspectives, perspectives which might colour one's observations and analysis, which might interfere with communication (an atheist that uses the word "peak experience" might not like talking to a theist who uses the term "mystical experience"), and which might deter some scholars from interest and investigation. There are, after all, large parts of the academy where the serious discussion of mystical experience is anathema (Ecklund and Long 2011). As I have found, it is a lot easier to get a colleague to talk about connection experiences than it is to get them to talk about mystical experience or transcendent experiences, which are often dismissed out of hand. When we use those words, the blinders go up and the resistance comes out. More neutral terms, terms which do not imply ontological leanings and which can incorporate materialist, theist, and atheist perspectives, avoid that resistance, facilitate communication between scholars of various bents, and encourage wider interest in the phenomenon. Using these agnostic terms allows one to push aside intractable ontological and phenomenological differences and concerns,

at least temporarily, while we open discussion, analyze the phenomenon, and try to sort the field out. Put another way, with these terms, we can talk about the phenomenon without worrying too much about what exactly one is connecting to. Thus the terms can refer, depending on one's ontological predilections, to connection to deeper neurological structures and functions of the brain (Carhart-Harris and Friston 2010; Garrison et al. 2015; Newberg et al. 2001) or, if one is willing to consider that consciousness might be rooted outside the body, to wider realities—e.g., a "numinous order" (Otto 1917), "extended mind" (Jahn and Dunne 2009), **Fabric of Consciousness (Sosteric 2016)**, or non-local consciousness beyond the body (Dossey 2015).

Finally, a third reason for using these terms is that they facilitate, in my view, the construction of a coherent conceptual framework that allows for clear descriptions of various aspects of the field, tighter conceptual integration and, consequently, much more fluid and transparent communication between scholars, perspectives, and disciplines. This becomes obvious when we consider some additional terms we might use to talk about this area and reflect upon how easily they tie various traditions, practices, and beliefs together. Thus, **connection practices** are practices, like the Holotropic Breath techniques developed by Stanislav and Cristina (Grof and Grof 1990), shamanic drumming (Drake 2012), Yoga, Zazen (How To Meditate: Zazen Instructions 2018), vision questing (Broker 1983; Eliade 1989), prayer, visualization, affirmation, taking a calm, slow walk in nature, watching a sunset, and so on, designed to facilitate, strengthen, and help purify connection. **Connection outcomes** are the numerous (usually positive, but sometimes negative) emotional, psychological, phenomenological, and cognitive outcomes of connection experience identified in the literature, including such things as enlightenment, epiphany (Bidney 2004), Noesis (Hanes 2012), Gnosis, etc. **Connection appliances** are material items like sacred stones and crystals (Harner 2013), spirit lodges and guardian boards (Deloria 2006), sweat lodges, and archetype decks (Sosteric 2021), Tibetan singing bowls, and so on, designed to facilitate connection and connection experience. **Connection supplements** are substances like Cannabis, Psilocybin, Peyote, chloroform (Bucke 2009), nitrous oxide,[17] DMT, LSD, Ketamine, MDMA, etc., that facilitate and force stronger connection. Connection supplements are typically referred to as psychedelics (mind revealing) or entheogens ("God Containing"). **Connection Obstacles** are obstacles that block connection. These obstacles include emotional obstacles (or poisons/kleshas, in Buddhist terminology) like greed, anger, pride, egotism, and jealousy (Sankaracharya 2001; Smith 1994), the "sins" of Christendom (particularly the seven deadly ones), and, as evidenced by my own experiences, ideology and fear. Connection obstacles also include elite-seeded archetypes which cause fear and confusion, as well as psychological damaged accrued as a consequence of toxic socialization (Sosteric and Ratkovic 2016), both of which interfere with, corrupt, and even prevent connection flows. **Connection pathologies** are pathologies, like psychological or emotional breakdowns (Grof and Grof 1990), that can sometimes be caused by powerful connection experiences, particularly when they are underlying mental health issues, when set and setting are not properly attended to, or when one is filtering experiences through elite ideology and archetypes, as I did for my first experiences. **Connection psychosis** are serious connection pathologies defined by the uncontrollable dissolution of ego boundaries coupled with a usually temporary, rarely permanent, break with reality.[18] Finally, we can understand **Connection frameworks** as schools of thought and practice devoted to identifying connection obstacles, developing connection practices, utilizing connection appliances, and providing sacred (read connection) spaces (e.g., temples) for connection practices (like meditation) or the exploitation of connection supplements. Connection frameworks are designed to teach about, facilitate, and improve connection experiences and connection outcomes. Connection frameworks are an important area of study. They have existed for thousands of years, in Vedanta, in Buddhism, in Zen, in Sufi branches of Islam (Ernst 1999), in monastic Christian practices (Jantzen 1989, 1990), and even in modern scholarly settings where the experience is carefully contained and controlled in the interests of the accumulating classes (Bender 2010). In addition, various modern attempts have

been made to develop a consistent and sophisticated connection framework, including elite Theosophical (Blavatsky 1889) and Western esoteric (Cicero and Cicero 1996, 2004) attempts, as well as more grounded, grass-roots efforts (De Christopher 1982; Ichazo 1976; Starhawk 2011).

The proliferation of terms to describe the phenomenology and outcomes of connection experience is problematic. Collapsing all that into a single term "connection experience" allows one to integrate and discuss, as illustrated above, a wide variety of relevant phenomena in a neutral and agnostic fashion. Unfortunately, this conceptual collapse occludes the varied and complex phenomenology of the experience. With it, we lose the rich field of terms (e.g., Satori experience, Enlightenment Experience, Flow Experience) which refer to unique ontological or epistemological aspects of the experience. The solution to recapturing the complex phenomenology is to categorize and operationalize CEs along five neutral **Connection Axes (CAs)**. These five CAs are quality, intensity, duration, content, and outcome. Here, **quality** refers to the general quality of the experience, whether felt as a positive zenith experience or fearful, negative nadir experience. **Intensity** refers to the phenomenological intensity of experience, which can range from minor nature experiences (appreciating the power and glory of nature and our connection to it) to experiences of cosmic enlightenment and searing cosmic bliss. **Duration** refers to the temporal duration of the event. **Content** refers to cognitive/emotional content. Content can include anything from small personal insights to appreciations of nature and oneness to grand cosmological revelations to eschatological prophecy. Finally, **outcomes** are the psychological, emotional, physical, and spiritual outcomes of the event, including things like enlightenment, enhanced tolerance (Parish 1999), healing (Miller and Baca 2001), moral quickening (Bucke 2009), expansion of empathy and compassion (Grace 2000), and perhaps even Siddhis (Akhilananda 1948) or advanced spiritual powers. These five connection axes represent a comprehensive typology for categorizing and operationalizing connection experience, one that provides a neutral and agnostic ground and which can potentially capture the richness and complexity of the experience.

As we can see, the above framework provides a conceptually consistent ground for describing, operationalizing, and theorizing the phenomenon under question. We can see the utility of this scheme above in the way it allows us to tie things together into a coherent understanding, and we can see the utility when we use it to describe specific experiences. For example, we can use the nomenclature to describe my own initial connection experience as an intense, short duration, nadir experience with Christian eschatological content and with positive psychological and creative outcomes. Similarly, one might describe the connection experiences of a Buddhist monk who experiences Daigo (Dōgen 1966) as an intense, short duration, zenith experience with Buddhist spiritual content and with positive cognitive and emotional outcomes. In this case, we can see how using the schema allows us to talk about two different experiences from two different cultures in a conceptually tight and meaningful way.

*4.4. Stage Four: Autoethnographic and Reporting Stage*

The final stage of the research project is the autoethnographic and reporting stage. In this stage, I begin to report on my experiences and findings, both in traditional scholarly formats and in this autoethnographic format, a format I feel particularly suited to discussing the various phenomenological and cognitive aspects of connection experience. Having published several papers examining the phenomenology of the experience (Sosteric 2016), their ubiquity (Sosteric 2018a), their progressive and revolutionary potential (Sosteric 2018b), and elite impositions on religious institution sand spiritual narratives (Sosteric 2014, 2020b), I am well into this stage, though nowhere nearing completion. This paper continues the reporting by providing a broad overview of themes identified in the research program and by linking these themes to my own research and, where possible, psychological and sociological work. This paper also provides the first autoethnographic account of my experience, with an emphasis on linking elements of this experience to subsequent conceptualization

and theorization. This paper does not dive to deeply in the "mystical" elements of the experience, elements which I dealt with elsewhere and that I plan to deal with in more detail in future work, but stays focused on the psychological and sociological elements of the phenomenon, with an emphasis on explicating the psychological, emotional, and scholarly challenges I experienced, as well as the proposed methodological and conceptual solutions to these challenges (staged data collection and analysis, using a MediaWiki to record, providing an agnostic nomenclature, etc.). I believe, given the dearth of both socio-logical analysis and scholarly accounts of connection experience, this focus is necessary and useful; however, I will leave it up to the reader to decide.

## 5. Discussion and Conclusions

As noted in the introduction, this paper takes a braided approach to telling the autoethnographic story of my connection experiences and the research and analysis that ensued. In particular, the paper attempts to braid together psychological, sociological, methodological, and multidisciplinary chords into an acceptable contribution that advances our understanding of connection experience. Each braid, though thin because of space limitations, highlights certain relevant issues. Each braid points to various avenues for further research and analysis.

Psychologically, the paper provides a case-study confirmation of many of the psycho-logical and emotional aspects of the experience reported in the literature, specifically their powerful, transformative, effusive, growth inducing effects, as well as their emotionally, existentially, and professionally challenging nature. More interesting perhaps, the paper also points, briefly, in the direction of elite religious ideology and archetypes and the fears, distortions, and blockages these introduce. Archetypes are certainly a concern of some psychologists, particular Jung and his followers. While Jung accepted archetypes as natural outgrowths of our evolutionary development (Jung 1980), this paper, which highlights elite interference in religious narratives and archetypes, points toward a need to take a more critical stance on these archetypes and religious narratives in order that we might better understand their impact on our individual and collective consciousness and psychology.

Sociologically, the paper addresses an existing, but insufficiently explored, sociological braid concerning elite interference in the religious institutions and spiritual/mystical narra-tives of this planet. This paper extends concern and highlights the potential significance of this interference. In this regard, the paper points to two potentially fruitful avenues of theoretical exploration, Ruyle's (1975) schema for theorizing this interference, a schema which casts religions as ideological institutions that elites use to maintain and reproduce what he calls their Regime's of Accumulation, and the French Structuralist school, particularly Foucault, for understanding the nature and significance of their interference as they colonize and shape this world's religious institutions and spiritual narratives. There has been some work in this area (Bender 2010; Boyce 2001; Decker et al. 1996; Jantzen 1995; Sosteric 2014, 2020b, forthcoming; Versluis 2007), but more needs to done, and quickly I feel.

Methodologically, the paper proposes novel methods that scholars who are interested in autoethnographic exploration of connection experience can use to manage and analyze the copious information flow that comes with connection. This included a staged research strategy designed to facilitate accurate recording of information flows and, later, grounded analysis and reporting, as well as the recommended use of MediaWiki software to record and analyze autoethnographic data. These methods are most relevant to scholars seeking to engage in autoethnographic analysis of connection experiences, but there may be elements useful to others, like the use of MediaWiki software as a sort of online scholarly notebook, or the initial inductive approach (collecting as much data as possible, and then doing the research and analysis). There were, of course, limitations in research methodology. One can think of alternate stages and alternate approaches to recording, analyzing, and researching the flow which might be equally fruitful. Perhaps one could start deductively, going to the literature first and then pursuing the experience. Certainly, a better use of MediaWiki software for this project could be envisaged. Given the opportunity to do the project again,

I would do some things differently. I would approach recording in the MediaWiki with a clear strategy for what to record and what to preserve. I would also recommend recording the specific reasons for specific edits.

Finally, there is a multidisciplinary braid. This braid is focused on the recognized problem of nomenclature confusion (Johansson 2022), a problem which makes understanding, analysis, and multidisciplinary communication a challenge. This paper offers one possible solution to the general confusion in the development of an agnostic and culturally neutral framework. The framework developed and reported here revolves around casting mystical experience as neutral connection experiences, where connection can either be to neurological systems or to wider realities of consciousness, and then developing a descriptive nomenclature to describe and operationalize various psychological and sociological aspects of this connection. A glossary of terms used in this paper is provided at the end. The utility of this framework was briefly demonstrated by using it to describe my experiences and the typical experiences of a Zen Buddhist monk, bringing both experiences closer and showing the various ways they are linked together.

At the end of this paper, what are we left with? This is a complex braid, so that is a tough question to answer, particularly because, I am sure, different people will get different things from this autoethnography. However, if I could specify what I wanted the reader to take away from this paper, there would be several things. First, I would want the reader to understand connection experiences are real, ubiquitous in the history of our species, and powerful, both in the sense of being powerfully healing and transformative and powerful in the sense of being potentially hard to handle. I would also want the reader to understand, as I have come to understand after exploring them for close to two decades, that there is a deep psychology to these events that needs to be carefully examined and that this psychology has been manipulated by elites who have colonized religious institutions and spiritual narratives in order to castrate connection experiences and propagate their own agenda. I would also want the reader to understand that until we accept that elites have been mucking around in this area for thousands of years, and that until we examine their inference in considerably more detail, we are going to have a hard time sorting things out in anything other than a simplistic and caricatured way.

Speaking of sorting things out, I would also, finally, want the reader to put aside ontological and cultural predilections and the confusion of terms and concepts these generate and instead embrace a more neutral and agnostic schema, one more suited to neutral communication, understanding, and analysis. This way, Buddhists who experience Satori, Christians who experience mystical marriages, Wiccans who connect with their Goddess, and atheists who tune into nature through peak experiences can all communicate and talk about these experiences in a way that heightens our understanding rather than enhancing our confusion. I suggest the schema provided in this paper, one that casts these experiences as connection experiences and then builds a lexicon up from there. The agnostic conceptual schema provided in this paper may allow us to start talking about these experiences in a more consistent, neutral, and operational fashion. This conceptual schema may also, because of its neutrality, encourage more open personal and scientific exploration. I think this is important because these experiences are incredibly important, not only personally but culturally, politically, and ecologically as well. As I have noted elsewhere, these experiences have the potential to transform the human race and potentially save the planet (Sosteric 2018b). Whatever we can do to open a more logical and grounded discussion, and whatever we can do to encourage sensible, effective, personal and scholarly exploration, will only benefit us and this planet in the long run.

**Funding:** This research received no external funding.

**Data Availability Statement:** Not applicable.

**Conflicts of Interest:** The author declares no conflict of interest.

### Glossary

| | |
|---|---|
| **Bodily Ego** | The bodily ego is your body's ego. It is the neurologically rooted ego that arises as a consequence of the operation of the brain's **Default Mode Network**. The bodily ego is functionally equivalent to Freud's conception of the ego as the center and source of your identity. |
| **Clearing Experience** | A clearing experience is a special sub-type of connection experience that leads to an abrupt and dramatic clearing of emotional, intellectual, psychological, or spiritual blockages to connection. A clearing experience, which can be either positive zenith or negative nadir, typically results in improved insight, understanding, health, well-being, and connection. |
| **Connection Appliances** | Connection appliances are material item like sacred stones and crystals (Harner 2013), spirit lodges and guardian boards (Deloria 2006), sweat lodges, archetype decks (Sosteric 2021), and so on, which facilitate connection and connection experience. |
| **Connection Axes** | Connection axes refer to five neutral axes along which we can characterize and describe connection experience. The five connection axes include quality, intensity, duration, content, and outcome. |
| **Connection Experience** (CE) | A connection experience is a discrete, short-term psychological, emotional, and physical experience of Connection that is sufficiently above one's average daily phenomenological experience as to be perceived by the individual as a qualitatively different state of awareness, consciousness, and being. The term connection experience is another name for a mystical, religious, or even peak experience, so-called because it represents a connection to either deeper neurological states or a wider reality of divinity and Consciousness (Dossey 2012, 2015). |
| **Connection Frameworks** | Connection frameworks are schools of thought and practice devoted to identifying connection obstacles, developing connection practices, utilizing connection appliances, and providing sacred spaces for the exploitation of connection supplements, all with a view towards understanding and facilitating CE and improving connection outcomes while at the same time minimizing connection pathology. |
| **Connection Pathology** | A connection pathology is a psychological/emotional alteration or breakdown of the bodily ego caused by a connection experience of an intensity, duration, quality, or content that an individual is not emotionally or psychologically prepared for. |
| **Connection Practices** | Connection practices are spiritual practices, like ancient yogic breathing practices (Akhilananda 1948; Brahmananda 1933), modern innovations in breath work (e.g., Holotropic Breathwork (Grof and Grof 1990)), drumming (Drake 2012), Yoga, Zazen (How To Meditate: Zazen Instructions 2018), vision questing (Broker 1983; Eliade 1989), prayer, taking a walk in nature, watching a sunset, and so on, which facilitate, strengthens, and helps purify connection and connection experience. |
| **Connection Psychosis** | A connection psychosis is an uncontrollable dissolution of ego boundaries coupled with a usually temporary, rarely permanent, break with reality. For an example of a permanent break with reality, see Schreber (2000). |
| **Connection obstacles** | Connection obstacles, which are any psychological, emotional, conceptual, or spiritual thing that interferes with, corrupts, or diminishes the flows that occur during connection events, are identified in both ancient and modern literature. |

| | |
|---|---|
| **Connection Supplements** | Connection supplements are substances like Cannabis, Psilocybin, Peyote, chloroform, nitrous oxide, DMT, LSD, Ketamine, MDMA, etc., that facilitate and force stronger connection. Connection supplements are typically referred to as *psychedelics* (mind revealing) or *entheogens* ("God Containing"). |
| **Ego Inflation** | Ego inflation is a type of connection pathology. It is an inflation of ego that leads to delusions of grandeur and importance. Ego inflation may be a psychological/emotional compensation for feelings of inferiority and exclusion. |
| **Flow Control** | Flow control refers to the ability to *control* the thoughts and images that flow through the mind during powerful connection events, particularly the negative, judgmental, ideological thoughts and emotions that can sometimes attend powerful and intense CEs. |
| **Nadir Experience** | A nadir experience is a negatively felt Connection Experience. Nadir experiences are unpleasant moments of stress, anxiety, anger, confusion, fear, paranoia, and even psychosis caused when Connection occurs and the individual is unprepared, damaged, embedded in a toxic milieu, or filled with ideologically rooted Wrong Thought. |
| **Normal Consciousness** | Normal consciousness is the common phenomenological experience of our normal, everyday waking state of state of consciousness. It is what we feel like when are not *lifted* into an alternate state. |
| **Spiritual Ego (theoretical)** | The Spiritual ego is that part of your identity in non-local mind (Dossey 2015) **or the Fabric of Consciousness (Sosteric 2016)** |
| **Zenith Experience** | A zenith experience is a positively felt mystical experience. These may range in power from minor nature and peak experiences to full-blown visionary revelations. |

## Notes

[1] A nadir experience is a negatively felt Connection Experience. Nadir experiences are unpleasant moments of stress, anxiety, anger, confusion, fear, paranoia, and even psychosis caused when Connection occurs and the individual is unprepared, damaged, embedded in a toxic milieu, or filled with ideologically rooted Wrong Thought. As far as I can tell, the first use of the term Nadir Experience to describe the quality of CEs was in 1963 (Thorne 1963, p. 50).

[2] A clearing experience is a special sub-type of connection experience that leads to an abrupt and dramatic clearing of emotional, intellectual, psychological, or spiritual blockages to connection. A CLE typically results in improved insight, understanding, health, well-being, and connection.

[3] A connection experience is a discrete, short-term psychological, emotional, and physical experience of connection that are sufficiently above one's average daily phenomenological experience as to be perceived by the individual as a qualitatively different state of awareness, consciousness, and being.

[4] My experiences cover a range of rather standard phenomenological experiences and psychological and emotional outcomes identified by Hood (Hood 1975; Hood et al. 2001) and others, and associated with connection experiences, including experiences of noesis, joy, happiness, bliss, oneness (Harmless 2008; Miller 2004), peace and contentment (Bourque and Back 1968; Laszlo et al. 1999), "enlightenment" (Neher 1990) and so on.

[5] At first glance, invoking decolonization may seem questionable; but as we see in the main body of this paper, it is quite apropos when we discuss spiritual topics because religious institutions and spiritual narratives are colonized and contested spaces. If we want to truly understand religious institutions and spiritual narratives, we have to acknowledge that colonization and work to understand it. Otherwise we, and by we I mean researchers and scholars, will simply reproduce colonized understandings, colonized narratives, and colonized research projects.

[6] Connection supplements are supplements (like Cannabis, Psilocybin, Peyote) or substances (like DMT, LSD, Ketamine, MDMA, etc.) that facilitate and force stronger Connection to Consciousness. Connection Supplements are typically referred to as *psychedelics* or *entheogens* ("God Containing"). Since the action of entheogens is to open a Connection to The Fabric of Consciousness, or to deeper neurological phenomenon, Connection Supplement is the superior term. https://SpiritWiki.lightningpath.org/index.php/Connection_Supplement (accessed on 5 September 2022)

[7] There is a tendency among scholars and others to distinguish between "psychedelic" experiences and "authentic" mystical or spiritual experiences, as if the imbibing of a substance necessarily invalidates or calls into question the validity of the experience. Although it is too soon to draw any conclusions, research into the neurological impact of connection supplements on brain function (Carhart-Harris et al. 2012; Carhart-Harris and Friston 2010; Hasenkamp and Barsalou 2012), coupled with similar research on the impact of meditation (Brewer et al. 2011; Farb et al. 2007), indicate that connection supplements modify brain

neurology in the exact same way as meditative practices. In addition, connection supplements like Peyote and Ayahuasca have been long used in Indigenous connection rituals (Smith 1964), and some religious officials admit they provide authentic experiences (Saunders 1995). There have been some statements in this regard. For example, "When the current philosophical authority on mystical experience, W.T. Stace, was asked whether the drug experience is similar to the mystical experience, he answered 'It's not a matter of it being similar to mystical experience; it is mystical experience". (W.T. Stace quoted in Smith 1964, pp. 523–24). Indeed, jumping over a fifty year dry spell (resulting from the American "war on drugs") which made research into any aspect of entheogen experience illegal, scholars are once again beginning to suggest there is a link between entheogens and authentic mystical experience (Ellens 2014).

8    The majority of CEs are **Zenith Experiences**, or experiences that are positive, life affirming, and healing (Bien 2004; Miller 2004). Not all CEs are zenith experiences, however. Some can be negative, fearful, paranoid, even schizophrenic "dark night of the soul" events. Little is known about these nadir experiences because most researchers ignore them or discount them as invalid. However, a few things can be said. Most nadir experiences are transitory and individuals inevitably recover, as I did. Sometimes, nadir experiences can lead to growth and transformation (Forer 1963). Sometimes, especially in situations where the individual's psyche is badly damaged, a permanently disordered connection may result. See for example, Memoirs of my Mental Illness (Schreber 2000), a case study of an individual with a disordered connection.

9    Madame Blavatsky, a member of Russian royalty, a famous New Age mystic, and author of the two volume theosophical treatise The Secret Doctrine, and Isis Unveiled said her mystical experiences gave her knowledge and information of things she had never studied (Kuhn 1930).

10   The term connection practice refers to the regular and disciplined daily practice of connection. Connection practice includes not only the actual practice of connection via the use of connection techniques like meditation, any preparatory work and study required to expand understanding, but also any healing practice required to heal psychological or emotional damage, and also any cognitive and psychological practices (time for self-reflection, etc.) required to ground and strengthen one's connection experiences.

11   Note, the presence of any fear can preempt an emerging connection and frighten one away from ongoing exploration. Church instilled fears of a judging and punishing God had suppressed my spiritual facilities and frightened me away from exploration for many decades. Other fears work the same way, that is, to suppress nascent connection. One person I spoke to, for example, had several short experiences, reporting notable phenomenological shifts accompanied by sudden intense flows of unfamiliar ideas. Unfortunately, the shift and intense flow frightened them every time. As a consequence, they shut down their nascent connection and eventually gave up their connection practice, never progressing past the momentary experiences. Given the prevalence of various fears in the general population, I expect this is a common story among many.

12   **Flow Control** refers to the ability to *control* the thoughts that flow through the mind, particularly the negative, judgmental, ideological thoughts and emotions that can sometimes attend powerful and intense CEs. Flow control may be exercised with pure will, by simply setting aside unwanted thoughts. If simple will is insufficient, deep breathing, affirmation and visualization can help distract and redirect thoughts. For example, if during connection you experience feelings and thoughts of unworthiness, press these thoughts and feelings away, perhaps with the help of an appropriate affirmation, something like "I am worthy, I am strong. I am worthy, I am strong." Note that flow control is not about stopping the flow of thoughts, as in some practices. It is about keeping ideas that would otherwise cause fear, confusion, and blockage at bay. Also note, you do not necessarily have to push away all negative thoughts. In some cases, you may wish to *lean in* to the thought patterns so that you can become aware they exist, and so you can assess their impact on you once you return to normal consciousness. This is true if the negative thoughts represent aspects of your life that need to be attended to, like if there is a pedophile in your family, or you are being violent towards your children, or other people. Those negative thoughts need to be accepted into the flow, acknowledged, and then action needs to be taken to address the negative behaviours.

13   These enhancements are identified in the Christian corpus as fruits of the spirit. From Galatians 5: 22, "But the fruit of the Spirit is love, joy, peace, forbearance, kindness, goodness, faithfulness, gentleness and self-control." In the Vedic corpus they are referred to as Siddhis, "Yoga powers are forms of extraordinary knowledge, such as awareness of previous rebirths, knowing the minds of others, seeing distant and hidden things, and remarkable abilities such as the power to become invisible, enter others' bodies, fly through the air, and to become disembodied for a period of time, which are traditionally thought to be attained as yogins progress in their practice" (Jacobson 2012).

14   Early and perhaps still influential attempts to demarcate mystical experience, for example, saw men rejecting the emotionally effusive mystical experiences of people like St. Teresa as "unpalatable," unbalanced, "hysterical emotional . . . " " . . . weakness, and not part of the "core of mystical experiences" (Stace 1960, pp. 51–53).

15   See https://SpiritWiki.lightningpath.org (accessed on 10 November 2021).

16   A connection pathology is a psychological/emotional alteration or breakdown of the bodily ego caused by a CE of an intensity, duration, quality, or content that an individual is not psychologically or emotionally prepared for.

17   Nitrous oxide has been used by Osho (Milne 2015), psychologist William James (James 1903, 2009), and others (Huston 2000) to facilitate CE.

18   For an example of a permanent break with reality, see Schreber (2000).

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
