# Peer review of "Connection 100—An Auto-Ethnography of My (Mystical) Connection Experiences"

_religions, doi:10.3390/rel13100993_

Round 1

Reviewer 1 Report

The author's account of mystical experience is too linked to a emotional framework and lacks a metaphysical insight. The very core of a 'mystical' experience, specially if understood as a 'connection experience' is the 'object' to which one is connected: this transcendent dimension is the more important one for many mystics so that some of them such as John of the Cross or Therese of Avila think that a supposed 'mystical experience' should be proofed and judged. For them the emotional impact does not define the experience but the object to which one is connected. Therefore makes much sense to deny the very idea of an atheistic mystical experience.

On the other side, the most arbitrary assumption of the article is the reduction of religious institutionalization to a ideological control tool. The author does not provide enough evidence to support this assumption and any argument against the elements that contradict such idea is alleged: such elements are, for example, the great amount of religious leaders that were ready to suffer and even dead to defend their creed, that many of them were also 'mystics' and not only 'rulers', etc.

Author Response

The author's account of mystical experience is too linked to a emotional framework and lacks a metaphysical insight. The very core of a 'mystical' experience, specially if understood as a 'connection experience' is the 'object' to which one is connected: this transcendent dimension is the more important one for many mystics so that some of them such as John of the Cross or Therese of Avila think that a supposed 'mystical experience' should be proofed and judged. For them the emotional impact does not define the experience but the object to which one is connected. Therefore makes much sense to deny the very idea of an atheistic mystical experience.

Fair points, but this paper is not addressing the metaphysical side of things, it is focused on using my own experiences to develop a sociological framework for understanding connection experiences. I have added a statement in the abstract to highlight this.

On the other side, the most arbitrary assumption of the article is the reduction of religious institutionalization to a ideological control tool. The author does not provide enough evidence to support this assumption and any argument against the elements that contradict such idea is alleged: such elements are, for example, the great amount of religious leaders that were ready to suffer and even dead to defend their creed, that many of them were also 'mystics' and not only 'rulers', etc.

Though it does have this element, I do not intend to reduce religion to an ideological tool. I have added a note to this effect. See footnote 13

Reviewer 2 Report

116 — Comma, not a semicolon

205 — Were, not where

235 — Phenomena, not phenomenon 

266 — Question mark, not a period

Footnote 11 — Missing a quotation mark on the bottom line 

324 — Correspond, not corresponded

347 — “…Straightforward to track simply be examining…” is awkward phrasing 

399 — Indoctrination is capitalized 

439 — “ You find this [Zoroastrian] framework in traditional religions like Judaism…”  How can that be when Judaism predates Zoroaster?  

444 — Canon, not cannon

468 — Free rein, not reign

667 — Blavatsky is bold

I feel very bad the author thinks religion is (almost) nothing but a power struggle for elitism (443-459, 472-485).  I also do not understand the comment about “[sociologists’] failure [to engage with religious artifacts] has allowed elites free reign to manipulate this planet’s spiritual discourses” (468-469).  Sounds like sociology would save the world?  

“[This failure] has even facilitated the uncritical importation of elite archetypes into Hegelian and Marxian theory” (469-470).  Hegel and Marx already were elitists, but I think the author is trying to say that their theories do not need to be.  

While I am not an expert on the world’s religions, it’s also disconcerting the author has not considered the possibility (at least here) that the Catholic Church actually knows something about the other side, and so forbids certain spiritual practices to protect her people, not keep them under control.  The author gets the control factor right as regards occultic practices.  

Author Response

I feel very bad the author thinks religion is (almost) nothing but a power struggle for elitism (443-459, 472-485). 

I don't think that at all. See my draft paper "The Sociology of Religion: A Decolonizing Approach." I do however think that the "power struggle" has been pretty much neglected by sociologists in favour of a weird fetishism with the ecclesiastic component of religion.

I have added a footnote, (13). to this effect

I also do not understand the comment about “[sociologists’] failure [to engage with religious artifacts] has allowed elites free reign to manipulate this planet’s spiritual discourses” (468-469).  Sounds like sociology would save the world? 

Not intented to be this. I've added a footnote (footnote 14)  which I hope provides an adequate explanation for this comment.

“[This failure] has even facilitated the uncritical importation of elite archetypes into Hegelian and Marxian theory” (469-470).  Hegel and Marx already were elitists, but I think the author is trying to say that their theories do not need to be. 

Not saying anything about the elitism of Hegel or Marx, just pointing out how religious dogma filtered into these theories.

While I am not an expert on the world’s religions, it’s also disconcerting the author has not considered the possibility (at least here) that the Catholic Church actually knows something about the other side, and so forbids certain spiritual practices to protect her people, not keep them under control.  The author gets the control factor right as regards occultic practices.

I have considered this. The Catholic church does know something about spiritual practices. This is why they have nunneries and monasteries where they can properly "contain" these experiences.

Reviewer 3 Report

These comments seem consistent with the goal of the special issue. I do wonder if the comments on the limitations of this approach as well as the obvious questions about generalization are worth reconsidering. I can't help but wonder if the author might want to reconsider some of the comments given the current zeitgeist, etc. Are there comments in here that will bother you years from now? Are their statements that your children will rib you about in the decades ahead? Although I admire the disinhibited approach, I would feel disheartened if I did not give you the chance to revise some of the comments before they appeared in print. 

Author Response

I would like to thank the reviewer for their concern for my future mental and emotional well-being and the admiration they express for my "disinhibited" approach. After going through each reviewer's comments and making requested changes, I am happy with the way I said things, do not feel I will be "bothered" by my comments in a few years, nor that my children will "rib" me in any manner.

Reviewer 4 Report

The article, in my opinion, has a clear idea, but the distinction between the Clearing Experience, the Zenith Experience is not clearly presented, and the connection between these phases and the cited scientific literature is not clearly demonstrated - thus showing the difference between the author's general level of expertise and his paper. Otherwise, I consider the article good enough to publish - after some fine editing and revision of the language - the author's language seems too informal at times.

Some concern arises from the frequent references to LSD, Cannabis, peyote, psilocybin - which may sound a bit like promotion.

Author Response

I have cleaned up the language and tried to make it less formal in places.

As requested, I have refined and improved definitions for Zenith and Nadir experiences, which are now clearly defined, on page 1.

On the topic of citation and expertise, the concept "nadir experience" is linked to the literature in several places throughout the paper. I even point out when the term was first used on page one, footnote 1. As far as I can tell, I am the first to use the term Zenith experience to describe positive connection experiences.

There are not many references to connection supplements (i.e., LSD, peyote, etc.) in the paper. Only two, one on page 2, and another p 13-14. This does not seem unreasonable given the significance of these substances in the history of human societies.

Round 2

Reviewer 1 Report

The author did not reconsider his/her main statements which I find highly controversial  and not enough supported, although they belong to the mainstream Marxist approach to religious institutions. To think that every religious structure such a church should be seen from the perspective of power is highly reductive and does not fit with the experience if one considers the history of the religions with a little objectivity and a view of its whole. It is not the question that the author thinks that religion is power. I see that it isn't true. The question is that the "church" (or analogous) are reduced to that.